# OmniResponse: Online Multimodal Conversational Response Generation in Dyadic Interactions

**Cheng Luo[1], Jianghui Wang[1], Bing Li[1]\*, Siyang Song[2], Bernard Ghanem[1]**
[1]King Abdullah University of Science and Technology, [2]University of Exeter

## Abstract

In this paper, we introduce Online Multimodal Conversational Response Generation (OMCRG), a novel task designed to produce synchronized verbal and non-verbal listener feedback online, based on the speaker's multimodal inputs. OMCRG captures natural dyadic interactions and introduces new challenges in aligning generated audio with listeners' facial responses. To tackle these challenges, we incorporate text as an intermediate modality to connect audio and facial responses. We propose OmniResponse, a Multimodal Large Language Model (MLLM) that autoregressively generates accurate multimodal listener responses. OmniResponse leverages a pretrained LLM enhanced with two core components: Chrono-Text Markup, which precisely timestamps generated text tokens, and TempoVoice, a controllable online text-to-speech (TTS) module that outputs speech synchronized with facial responses. To advance OMCRG research, we offer ResponseNet, a dataset of 696 detailed dyadic interactions featuring synchronized split-screen videos, multichannel audio, transcripts, and annotated facial behaviors. Comprehensive evaluations on ResponseNet demonstrate that OmniResponse outperforms baseline models in terms of semantic speech content, audio-visual synchronization, and generation quality. Our dataset, code, and models are publicly available at `https://omniresponse.github.io/`.

## 1 Introduction

Generating realistic human conversational responses has substantial potential across numerous applications, spanning from human-computer interactions [39], immersive metaverse experiences [30], to mental health interventions [31]. However, human communication is inherently multimodal and complex. In face-to-face interactions, speakers convey their messages not only through spoken language but also through non-verbal cues, such as lip movements and facial expressions. Correspondingly, listeners provide multimodal responses consisting of verbal (e.g., audible affirmations or disapprovals) and non-verbal responses (e.g., subtle head nods). While considerable efforts [10, 67] have been dedicated to modeling text dialogue, particularly in language-based interfaces [35], modeling multimodal conversational interactions has been much underexplored.

In this paper, we explore a new task: learning to simultaneously generate verbal and non-verbal listener [2] responses in an online dyadic conversation setting, conditioned on the speaker's verbal and non-verbal inputs (see Figure 1). We refer to this task as Online Multimodal Conversational Response Generation. Although various audio-to-video generation methods (e.g. talking head generation [82, 84, 79]) have shown impressive performance, these methods focus on synthesizing visual content aligned with input audio signals, which ignores explicitly modeling multimodal conversational interactions. Recent studies [41, 47, 61] propose to generate facial reactions for a

---

\*Corresponding author.

[2]Previous studies [8, 23] defined a speaker–listener framework for dyadic interactions, in which the listener both attends to the speaker's utterances and provides verbal and nonverbal feedback.

39th Conference on Neural Information Processing Systems (NeurIPS 2025).

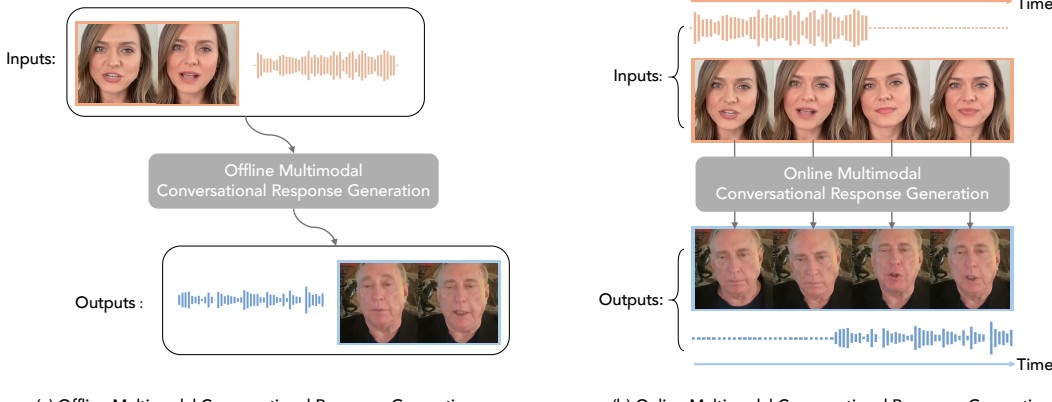

(a) Offline Multimodal Conversational Response Generation      (b) Online Multimodal Conversational Response Generation

Figure 1: **Illustration of the new OMCRG task.** (a) In offline tasks, the generation model generates the listener's full response only after receiving the entire input sequence from the speaker. (b) Differently, OMCRG task requires sequentially processing the speaker's incoming input and generating multi-modal responses for the listener on the fly.

listener; however, these methods overlook verbal responses, which are essential to engage in dialogue fully.

The OMCRG task is complex and poses major challenges in three aspects. First, it is non-trivial to directly achieve synchronization between the generated audio and facial reactions of the listener for OMCRG task. As revealed in existing talking-head works [82, 66], achieving precise alignment between facial motion and audio is already challenging, even when the entire audio signal is given. In contrast, OMCRG is to generate both audio and facial reactions simultaneously and incrementally. Such online and multimodal generation settings make face-audio synchronization much more difficult, due to the high variability and semantic ambiguity of audio modality. Second, due to the online setting, the model has to reason over partial speaker input and generate audio-visual responses on the fly, which requires both powerful audio-visual understanding and generation abilities. While powerful pre-trained models have been developed for language and vision, audio modeling remains comparatively underdeveloped, making it more challenging to generate expressive and appropriate audio and facial reactions. Third, the lack of high-quality datasets for dyadic multimodal interaction significantly hinders the development of OMCRG.

We address the above challenges by proposing a unified framework, OmniResponse, which autoregressively generates high-quality multimodal listener responses. Rather than directly synchronizing generated audio and facial reactions, our key insight is to introduce text as an intermediate modality for the OMCRG task. Compared with audio, text offers clearer semantics and reduces uncertainty, making it more tractable for learning multimodal reaction generation. However, text is a static modality without inherent temporal information, posing challenges for synchronizing spoken words with visual frames in an autoregressive generation setting. To overcome this, we introduce a Multimodal Large Language Model (MLLM) augmented with two innovative modules: Chrono-Text and TempoVoice. The Chrono-Text module temporally anchors generated textual tokens by incorporating additional tokens (markers) that explicitly encode time, ensuring alignment between words and visual frames. TempoVoice is a controllable, online text-to-speech module designed to produce synchronized audio from these temporally annotated textual embeddings, ensuring accurate synchronization between audio and facial reactions.

In addition, we construct a high-quality dataset named ResponseNet, comprising 696 dyadic conversation pairs. Each pair includes synchronized split-screen video streams of both speaker and listener, multichannel audio recordings, verbatim text transcriptions, and detailed facial-behavior annotations (*i.e.*, facial expressions and head movements). Through extensive retrieval for scarce dyadic video data, rigorous content filtering, meticulous camera-shift alignment, and manual annotation, ResponseNet delivers a unique and valuable resource for benchmarking OMCRG.

Our contributions are summarized as follows: (1) we present OmniResponse, the first online model to jointly process and generate synchronized streams of conversational human behavior, establishing a

foundation for future work in human–agent interaction; (2) we introduce ResponseNet, an annotated dyadic conversation dataset and benchmark, enabling standardized evaluation of OMCRG models.

## 2 Related Work

**Facial Reaction Generation.** Facial reaction generation (FRG) [63, 83, 60] is a particularly challenging new task as it requires to predict the non-deterministic human facial reactions under different contexts. Early FRG approaches [27, 28] relied on Generative Adversarial Networks (GANs) [43, 21] typically conditioned the generation process on the speaker visual-speech behaviors. Since FRG is a non-deterministic process (*i.e.*, different facial reactions can be triggered by the same speaker behavior [63]), recent advances have shifted towards more sophisticated generative frameworks. For example, Ng *et al.* [47] introduces a non-deterministic approach based on Variational Autoencoders (VAEs) [32], which enabled sampling diverse human facial motions. This work was complemented by a novel dataset containing paired recordings of active speakers and silent listeners, providing essential training data for modeling natural reactions. Zhou *et al.* [83] developed a specialized speaker-listener video dataset for head motion generation, which is somewhat limited by its relatively short clip durations (median length of 9.0 sec) and modest dataset scale (1.58 hours total), and thus constraining their model's ability to learn long-term temporal dependencies. More recent works have attempted to address these limitations through innovative architectural choices or larger-scale datasets [61, 62]. Luo *et al.* [41, 15] and Zhu *et al.* [85] proposed transformer-based [70] VAE and diffusion models [64, 24], respectively, training them on a hybrid collection of videos from three different human-human dyadic interaction datasets [12, 57, 50].

**Spoken Dialogue Models.** Spoken dialogue models generate natural speech responses in real-time, requiring systems to process both verbal content and paralinguistic elements of communication. Early approaches including AudioPALM [58], Spectron [46], and SpeechGPT [77] adopted pipelines combining automatic speech recognition (ASR), text generation, and text-to-speech (TTS) synthesis. However, their requirement to complete the entire response before the speech generation makes them unsuitable for live human-computer interactions. Recent developments [44, 18, 49] have shifted towards end-to-end approaches that directly model speech-to-speech generation. Representative examples include Moshi et al. [18] and dGSLM [49], which operate as full-duplex speech dialogue systems capable of processing continuous speaker input while generating appropriate vocal responses. While these advances are significant, they focus exclusively on speech and text modalities, overlooking the crucial visual aspects of human communication. Even recent work by Park et al. [51] that includes visual-speech data is limited to intermittent speaker-listener interactions.

**Autoregressive Generative Model.** Transformer-based autoregressive models [70] have revolutionized numerous domains in AI, demonstrating remarkable success in language modeling [10, 67], multi-modal processing [40, 3, 34, 45], and generative tasks [56, 81, 74, 73, 72, 65]. Their success can be attributed to their inherent scalability and ability to unify multi-modal training under a single autoregressive objective, enabling seamless integration of different data modalities. The adaptation of transformers to visual tasks was pioneered by approaches such as VQVAE [69] and VQGAN [19], which introduced effective methods for quantizing visual information into discrete tokens. They align visual generation with the successful paradigm of language modeling by employing decoder-only transformers to predict sequences of image tokens. Subsequent research [13] has focused on enhancing both the efficiency of tokenization processes [42, 37] and sampling procedures [76], while simultaneously scaling up model architectures to handle increasingly complex tasks.

## 3 Methodology

**Problem Definition.** Let $\mathbf{F}_t^s$ and $\mathbf{A}_t^s$ be the speaker's facial and audio cues at time $t$, respectively. Given the speaker's streaming facial sequence $\mathbf{F}_{1:t}^s$ and audio sequence $\mathbf{A}_{1:t}^s$ from time 1 to $t$, the goal of OMCRG is to online generate facial reactions $\mathbf{F}_t^l$ and audio feedback $\mathbf{A}_t^l$ at time step $t$. Such multi-modal generation has been much less underexplored, different from recent works [83, 41, 18, 77] mainly focusing on single-modal response generation. To provide natural responses, it is crucial to ensure that the generated facial reactions and audio are temporally synchronized and react appropriately to the speaker. However, this is significantly challenging due to the inherent difficulty of online audio-visual understanding and generation.

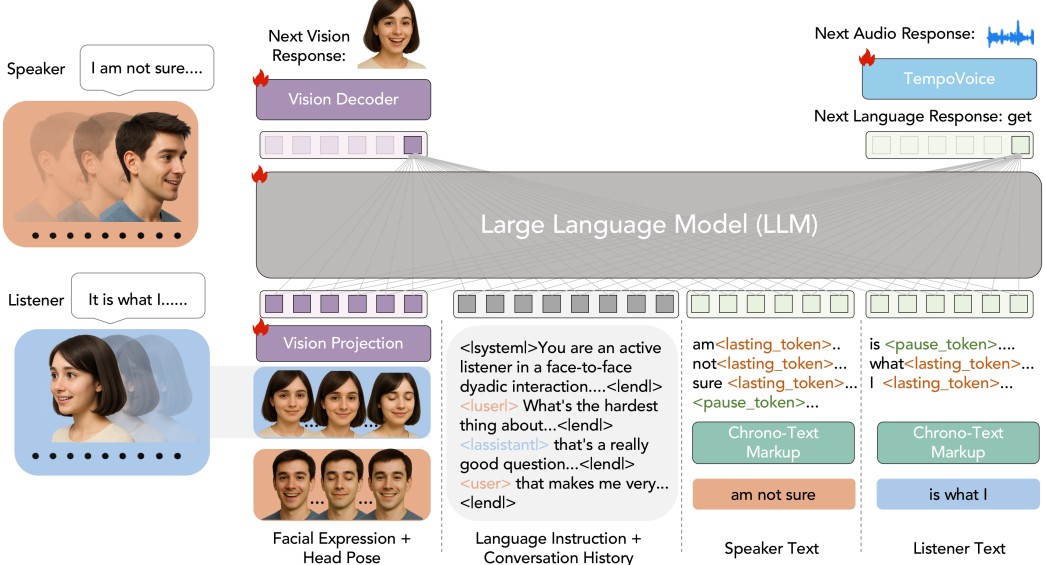

Figure 2: **Overview of the proposed OmniResponse**. The model takes textual conversational history and newly coming multimodal information (e.g., facial cues) from the speaker and listener as input, and generates temporally synchronized facial and textual responses for the listener by leveraging a pre-trained LLM enhanced with our proposed Chrono-Text Markup. The generated text embeddings are converted into audio synchronized with the facial response by the proposed TempoVoice module.

Instead of generating audio and visuals directly, we treat text as an intermediate modality and decompose OMCRG into two subproblems: (i) joint text–and–face response generation—producing temporally aligned facial reactions $\mathbf{F}_t^l$ and textual responses $\mathbf{W}_t^l$; and (ii) synchronous text-to-speech synthesis—converting $\mathbf{W}_t^l$ into audio waveform segments $\mathbf{A}_t^l$ that are aligned with the facial reactions. However, because text lacks explicit temporal information, achieving tight alignment with facial and audio streams is challenging for both subproblems. We address this issue with two novel modules.

**Overview.** We present OmniResponse, a novel framework for the OMCRG task (see Figure 2), where OmniResponse is a new MLLM enhanced by two proposed key components: *Chrono-Text Markup* and *TempoVoice*. In particular, our OmniResponse leverages the capability of a pretrained LLM to understand and interpret the speaker's multimodal inputs and autoregressively generate meaningful responses in terms of textual and facial responses. To address the lack of temporal information in text, the proposed *Chrono-Text Markup* embeds explicit temporal marks between text tokens, endowing the input and output text with time-aware embeddings and ensuring precise alignment with the generated facial reactions. Furthermore, the proposed *TempoVoice* generates audio responses temporally synchronized with both the generated textual response and the listener's facial movements.

## 3.1 OmniResponse

**Model Architecture.** As shown in Figure 2, OmniResponse processes multiple modalities from the speaker and the listener, temporally aligns different modalities, and outputs synchronous multimodal responses to the speaker. In particular, at each time step $t$, OmniResponse consumes: (1) *Static text inputs*: a task-specific instruction prompt $W_{\text{instruct}}$ and the conversation history prior to time $\tau$ ($\tau < t$), denoted $W_{\text{history},<\tau}$; and (2) *Temporal inputs*: the previously generated facial features of the listener $\hat{\mathbf{F}}_{\tau:t-1}^l$, the facial features of the speaker $\mathbf{F}_{\tau:t-1}^s$ and the accumulated text sequences from both participants ($\mathbf{W}_{\tau:t-1}^s$, $\hat{\mathbf{W}}_{\tau:t-1}^l$) over the interval $[\tau, t-1]$. Using these inputs, OmniResponse predicts the next facial features $\hat{\mathbf{F}}_t^l$, the verbal response $\hat{\mathbf{W}}_t^l$, and the corresponding speech segment $\hat{\mathbf{A}}_\mu^l$ in the current frame, ensuring precise temporal alignment in all modalities. Formally, we defined this process as:

$$\{\hat{\mathbf{F}}_t^l, \hat{\mathbf{A}}_\mu^l, \hat{\mathbf{W}}_t^l\} = \mathcal{M}\big(W_{\text{instruct}}, W_{\text{history},<\tau}, \mathbf{F}_{\tau:t-1}^s, \hat{\mathbf{F}}_{\tau:t-1}^l, \mathbf{W}_{\tau:t-1}^s, \hat{\mathbf{W}}_{\tau:t-1}^l\big).$$

**Vision Projection.** We introduce the vision projection layer to enable the pretrained LLM (Phi-3.5 mini-instruct with 3.8B parameters [1]) to process visual facial features. The layer is implemented as a multilayer perceptron (MLP) that maps the the listener's and speaker's past facial features $\hat{\mathbf{F}}^l_{1:t-1}$ and $\mathbf{F}^s_{1:t-1}$ into embedding features $\mathbf{V}_{1:t-1}$ aligned with the LLM token space. During autoregressive generation, the MLLM employs causal self-attention [70] to model temporal dependencies between the next token and previous one, and outputs the next listener vision embedding $\hat{\mathbf{V}}^l_t$.

**Vision Decoder.** A learnable vision decoder, comprising transformer layers, converts $\hat{\mathbf{V}}^l_t$ back into the original coefficient space to produce the predicted listener facial coefficients $\hat{\mathbf{F}}^l_t$. Subsequently, a pre-trained visual renderer maps these visual coefficients to 2D frames, using a given portrait image. Please refer to the appendix for additional details.

**Chrono-Text Markup.** Visual frames inherently encode temporal information, whereas text tokens are static and lack any temporal dimension. Additionally, visual frames and textual tokens typically differ in length due to their fundamentally different modalities, making unified autoregressive prediction challenging. To resolve this mismatch, we propose *Chrono-Text Markup*, a novel yet straightforward approach that explicitly embeds temporal information into textual data, aligning the textual sequence precisely with the visual frame sequence. Unlike prior approaches such as TimeMarker [14], which inserts timestamps only between visual frames or the method by Ng et al. [48], which integrates timestamp embeddings into textual tokens, our method employs only two special markers, ensuring that the textual and visual sequences have identical lengths. Specifically, we insert two special tokens into the transcript: [PAUSE] to denote silent intervals between utterances, and [LASTING] to indicate that the previous textual word continues speaking to the current time. Each text token is placed between pause and lasting tokens.

**Multimodal Context Modeling.** Our synchronous Multimodal LLM integrates both static and dynamic inputs: *Static inputs*: the instruction prompt and the accumulated conversation history. *Dynamic inputs*: frame-aligned visual embeddings and timestamped textual tokens for both speaker and listener. All tokens are jointly processed by an *omni-attention* mechanism that enforces causal, cross-modal interactions. Under this operation, each visual token attends to preceding visual tokens and to text tokens marked by chrono-text markers at earlier timestamps; similarly, each dynamic text token attends to past visual and textual tokens. However, this omni-attention prevents dynamic tokens from looking at future tokens. This ensures the generation adheres to temporal dynamics and cross-modal interactions. Meanwhile, static tokens remain globally accessible, ensuring that every dynamic update remains guided by the overarching instructions.

**TempoVoice.** Generating natural speech that is precisely synchronized with text and facial frames poses a significant challenge. To address this, we introduce a dedicated synthesis pipeline, *TempoVoice*.

Our framework begins by combining the listener's voiceprint, extracted via the Spark-TTS global tokenizer [71] to capture speaker identity, with the hidden states of the generated text (see Figure 3). We then apply sinusoidal positional encodings to the merged embeddings. Since audio-token sequences typically differ in length from visual frames and textual tokens, we prepend a series of zero-initialized placeholder tokens, each endowed with positional information. These placeholders serve as queries in a cross-attention module within a Transformer decoder, attending over the fused text–voice representations. This mechanism enables fully synchronous, autoregressive generation of audio tokens in lockstep with visual frames and text tokens. Finally, a linear projection layer maps the decoder outputs to logits over the discrete audio-codec vocabulary.

The decoder logits are then quantized into discrete audio semantic tokens $\hat{\mathbf{A}}_\mu$, as defined by the Spark-TTS audio tokenizer [71]. Conditioned on these semantics and the global speaker-identity embeddings, the tokenizer reconstructs the continuous waveform segment.

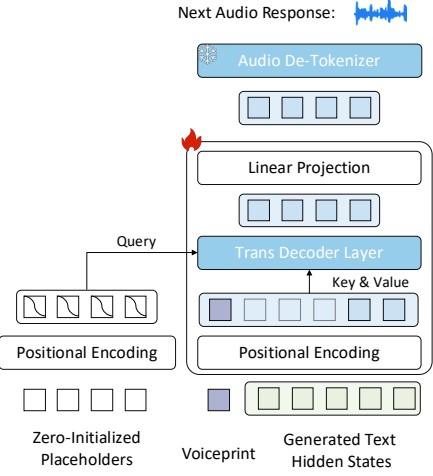

Figure 3: **Architecture of TempoVoice.** TempoVoice transforms textual hidden-state embeddings into audio segments.

## 3.2 Training Objectives

To train OmniResponse, the training objective is a weighted combination of text generation loss $\mathcal{L}_{\text{text}}$, vision reconstruction $\mathcal{L}_{\text{vision}}$, and audio generation loss $\mathcal{L}_{\text{audio}}$:

$$\mathcal{L} = \mathcal{L}_{\text{text}} + \lambda_{\text{vision}}\mathcal{L}_{\text{vision}} + \lambda_{\text{audio}}\mathcal{L}_{\text{audio}}, \qquad (1)$$

where $\lambda_{\text{vision}}$ and $\lambda_{\text{audio}}$ are the scaling factors balancing text, vision, and audio loss terms.

**Text Generation Loss.** The text loss encourages accurate next-token prediction conditioned on both speaker context and past listener states:

$$\mathcal{L}_{\text{text}} = -\sum_t \log p_\theta\big(W_t^l \ \big| \ W_{\text{instruct}}, W_{\text{history},<\tau}, \mathbf{F}_{\tau:t-1}^s, \hat{\mathbf{F}}_{\tau:t-1}^l, \mathbf{W}_{\tau:t-1}^s, \hat{\mathbf{W}}_{\tau:t-1}^l\big). \quad (2)$$

**Vision Reconstruction Loss.** To align predicted and ground-truth facial dynamics, we apply an $\ell_2$ reconstruction loss on the listener's feature embeddings:

$$\mathcal{L}_{\text{vision}} = \sum_t \big\|\hat{\mathbf{F}}_t^l - \mathbf{F}_t^l\big\|_2^2. \qquad (3)$$

**Audio Generation Loss.** The audio loss operates over discrete semantic tokens $\mathbf{A}_\mu^l$, indexed by $\mu$, which correspond to frame indices $t = \mu k$ ($k$ is the downsampling factor). We maximize the likelihood of each token conditioned on previous audio semantics and the listener's hidden states:

$$\mathcal{L}_{\text{audio}} = -\sum_\mu \log p_\theta\big(\mathbf{A}_\mu^l \ \big| \ \mathbf{A}_{<\mu}^l, \mathbf{H}_{t-k+1:t}\big), \qquad (4)$$

where $\mathbf{H}_{t-k+1:t}$ denotes the model's hidden representations for the corresponding listener text tokens $\hat{\mathbf{W}}_{t-k+1:t}^l$. This formulation ensures coherent alignment across modalities throughout generation.

## 4 Dataset Construction

Existing publicly available dyadic video datasets do not satisfy the requirements of the OMCRG task (See Figure 1). For example, mono-view talking-head datasets and offline dialogue corpora (e.g., MultiDialog [51]) do not offer split-screen recordings that capture speaker and listener simultaneously. Others, such as IEMOCAP [11], feature predominantly side profile views recorded in noisy environments and provide only mixed audio channels, thus preventing separate analysis of each participant's speech. Furthermore, datasets such as ViCo [82], ICD [47], and REACT2024 [61] lack comprehensive textual annotations, suffer from low video resolution [82, 11, 61], or exhibit inconsistent spoken languages [61]. To fill the dataset gap, we introduce ResponseNet that comprises 696 temporally synchronized dyadic video pairs, totaling over 14 hours of natural conversational exchanges. Each pair provides high-resolution ($1024 \times 1024$) frontal-face streams for both speaker and listener, along with separated audio channels to support fine-grained analysis of verbal and nonverbal behavior. Table 1 shows ResponseNet is the only dataset that satisfies the key requirements: (1) online video streaming, (2) separate audio channels, and (3) textual word-level annotations for both participants.

The construction of ResponseNet follows a rigorous workflow that integrates automated tools with extensive human-in-the-loop curation. (1) Initially, split-screen videos featuring simultaneous appearances of speaker and listener are sourced from YouTube according to predefined topic and quality criteria. These clips are then filtered to remove low-resolution, noisy, or frequent camera transitions. (2) Human annotators perform a thorough review to correct camera-view mis-alignments and ensure precise temporal synchronization between streams. (3) Next, mixed-channel audio tracks are automatically separated into discrete speaker and listener channels using speaker separation tools such as MossFormer2 [80] and subsequently verified and refined by experts. Finally, word-level transcripts are generated via automatic speech recognition [55] and meticulously proofread to guarantee accuracy. By combining automation with meticulous manual oversight across data sourcing, preprocessing, alignment, audio separation, and annotation, this pipeline yields a high-quality, richly annotated dyadic video corpus ideally suited for multimodal conversational response generation.

Table 1: **Comparison of conversation datasets.** 🧑 and 🧑 denote speaker and listener data respectively. ResponseNet provides complete multimodal data (speaker+listener) with their separated audios.

| Dataset | Video | Audio | Text | Online | Separated Audios | # Dialogues | Total Duration |
|---|---|---|---|---|---|---|---|
| MultiDialog [51] | 🧑+🧑 | 🧑+🧑 | 🧑+🧑 | ✗ | ✓ | 8,733 | 339.7h |
| ICD [47] | 🧑+🧑 | 🧑+🧑 | ✗ | ✓ | ✓ | 182,132 | 72h |
| ViCo [83] | 🧑+🧑 | 🧑 | ✗ | ✓ | ✗ | 483 | 1.6h |
| REACT2024 [62] | 🧑+🧑 | 🧑+🧑 | ✗ | ✓ | ✓ | 5,919 | 71.8h |
| IEMOCAP [11] | 🧑+🧑 | 🧑+🧑 | 🧑+🧑 | ✓ | ✗ | 151 | 11.5h |
| **ResponseNet** | 🧑+🧑 | 🧑+🧑 | 🧑+🧑 | ✓ | ✓ | 696 | 14.2h |

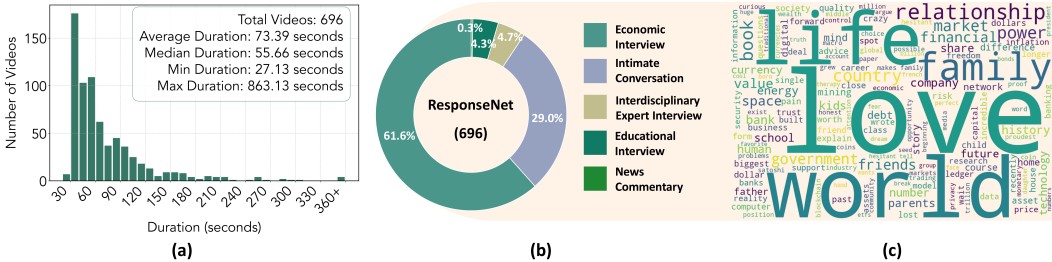

Figure 4: **Statistics of ResponseNet**. (a) Distribution of video clip durations. (b) Distribution of dyadic conversation topics. (c) Word cloud of spoken words in dyadic conversations.

The statistics of ResponseNet are shown in Figure 4. The durations of speaker-listener video clips range from 27.13 seconds (short conversations) to 863.13 seconds (long conversations) in ResponseNet. Figure 4.(a) shows that the average clip duration in ResponseNet is 73.39 seconds, significantly longer than that of other dyadic datasets such as REACT2024 (30 seconds), and ViCo (9 seconds). This extended duration ensures that each clip captures sufficient conversational exchanges. Figure 4.(b) illustrates that the conversations span a diverse range of topics, including professional discussions (e.g., economic interviews, news commentaries), emotionally driven interactions (e.g., intimate conversations), educational settings (e.g., teaching interviews), and interdisciplinary expert discussions. Figure 4.(c) presents a word cloud highlighting the most frequent words in the conversations. Such diversity shows that ResponseNet captures rich and varied human-human interactions rather than being restricted to narrow or monotonic conversation patterns.

## 5 Experiments

**Implementation Details.** Our framework was implemented using PyTorch [52] and trained on four NVIDIA Tesla A100 GPUs. The model optimization was performed using the AdamW optimizer [33] with a learning rate of $2 \times 10^{-5}$, $\beta_1 = 0.9$, $\beta_2 = 0.999$, and a weight decay of $10^{-4}$, accompanied by a cosine learning rate scheduler. Training was executed with a batch size of one for 2,000 epochs. Additionally, we fine-tuned the LLM using the LoRA [26] technique with a LoRA rank of 64 and a LoRA alpha value of 16. More implementation details are provided in the Appendix.

**Evaluation Metrics.** Quantitatively evaluating the quality of multimodal response generation remains non-trivial. We thereby employ comprehensive metrics to evaluate generation results across text, audio, and visual modalities. For text response, we use METEOR [9], BERTScore$_{F1}$ [78], and ROUGE-L [38] to measure how *appropriate* and *natural* the generated responses are, based on reference responses from the ResponseNet test set. We also adopt Distinct-2 [36] to evaluate *diversity* through the ratio of unique bi-grams. For audio response, we adopt UTMOSv2 [6], a neural MOS predictor that estimates the perceptual naturalness, and employ LSE-D [54, 16] (Lip–Speech Error Distance) to evaluate *synchronization* between generated speech and lip movements. For facial response, we compute Fréchet Distance (FD) [4] between real and generated facial-feature distributions, and Fréchet Video Distance (FVD) [68] to assess the spatial–temporal *visual quality* of generated video sequences.

Table 2: **Quantitative Results on ResponseNet test set.**

| Model | Text | | | | Audio | | Video | |
|---|---|---|---|---|---|---|---|---|
| | METEOR ↑ | BERTScore$_{F1}$ ↑ | ROUGE-L ↑ | Distinct-2 ↑ | LSE-D ↓ | UTMOSv2 ↑ | FD ↓ | FVD ↓ |
| Ground-Truth | – | – | – | 0.835 | 8.96 | 1.56 | – | – |
| *Offline Text Dialogue Generation System* | | | | | | | | |
| GPT-4o [2] | 0.167 | 0.805 | 0.079 | 0.928 | – | – | – | – |
| GPT-4 [2] | 0.163 | 0.822 | 0.082 | 0.960 | – | – | – | – |
| GPT-o1 [2] | 0.189 | 0.822 | 0.113 | 0.948 | – | – | – | – |
| Qwen-7B-Chat [7] | 0.167 | 0.807 | 0.090 | 0.920 | – | – | – | – |
| Claude-Sonnet-4 [5] | 0.183 | 0.807 | 0.101 | 0.966 | – | – | – | – |
| Gemini-2.5-Flash [17] | 0.175 | 0.824 | 0.085 | 0.932 | – | – | – | – |
| DeepSeek-R1 [22] | 0.173 | 0.815 | 0.078 | 0.981 | – | – | – | – |
| *Online Auditory Dialogue Generation System* | | | | | | | | |
| Moshi [18] | 0.120 | 0.818 | 0.078 | 0.499 | – | 2.21 | – | – |
| *Facial Reaction Generation System* | | | | | | | | |
| ReactFace [41] | – | – | – | – | – | – | 32.72 | 340.28 |
| ViCo [83] | – | – | – | – | – | – | 57.13 | 325.65 |
| *Online Multimodal Conversational Response Generation Baseline* | | | | | | | | |
| LSTM [25] | 0.042 | 0.716 | 0.000 | 0.000 | 9.72 | 1.21 | **6.51** | 320.92 |
| Audio-visual LLM | 0.030 | 0.662 | 0.020 | 0.155 | 10.03 | 1.32 | 580.86 | 681.55 |
| OmniResponse (Ours) | **0.141** | **0.806** | **0.081** | **0.882** | **9.56** | **1.41** | 15.46 | **314.94** |

## 5.1 Quantitative Results

To the best of our knowledge, few works have explored the OMCRG task before. We build two baselines and compare them in Table 2: (1) LSTM-based method employing a recurrent neural network [25] for temporal sequence modeling; (2) Audio-visual LLM taking speaker–listener audio and visual inputs and leveraging pre-trained LLM to generate audio–visual frames autoregressively. Table 2 further reports the generation performance of representative single-modality baselines, including offline, text-only dialogue models (e.g., GPT variants [2], Qwen-7B-Chat [7], Claude-Sonnet-4 [5] (version 2025-05-14), Gemini-2.5-Flash [17], and DeepSeek-R1 [22] (version 2025-05-28)), online audio-only generation models (e.g., Moshi [18]), and facial reaction generation approaches [41, 83]. Different from these methods focusing on generating a single modality, our method enables online, synchronized generation across audio, visual, and textual modalities for modeling human conversation.

Table 2 shows that our OmniResponse achieves the best performance in dialogue speech content (METEOR, BERTScore$_{F1}$, ROUGE-L, Distinct-2), audio quality (UTMOSv2), audio–visual synchronization (LSE-D), as well as temporal consistency and visual quality (FVD). Although the LSTM baseline achieves a lower FD owing to its tendency to produce repetitive static visual output, it fails to generate rich, synchronized multimodal responses. Audio-Visual LLM does not incorporate the text modality, compared to our method. Consequently, Audio-Visual LLM achieves much lower speech content quality (METEOR and BertScore$_{F1}$) and struggles with audio–visual synchronization (LSE-D) than our method. Although Audio-Visual LLM leverages a powerful LLM, it is still challenging to directly synchronize generated audio with facial reactions, especially in the absence of a strong audio foundation model.

Our OmniResponse model significantly outperforms Audio-visual LLM across all evaluated metrics, including non-verbal ones. These results demonstrate that introducing text as an intermediate modality greatly enhances the naturalness and realism of non-verbal responses, as reflected by the FD and FVD scores. Moreover, we introduce a novel framework that effectively adapts pre-trained LLMs for audio–visual generation with the proposed Chrono-Text Markup and Tempo Voice.

## 5.2 Qualitative Results

Figure 5 presents a qualitative result. The synthesized listener remains silent while the speaker is speaking, but then produces an immediate or delayed response at the end of each speaker turn. This behavior demonstrates that OmniResponse effectively captures the temporal dynamics of online dyadic conversation and generates responses at appropriate timestamps. For example, between 100.97 and 132.05 s, the listener interjects briefly between 120.13 and 121.57 s in response to the speaker's ongoing content, reflecting natural human conversational interaction. In contrast, a

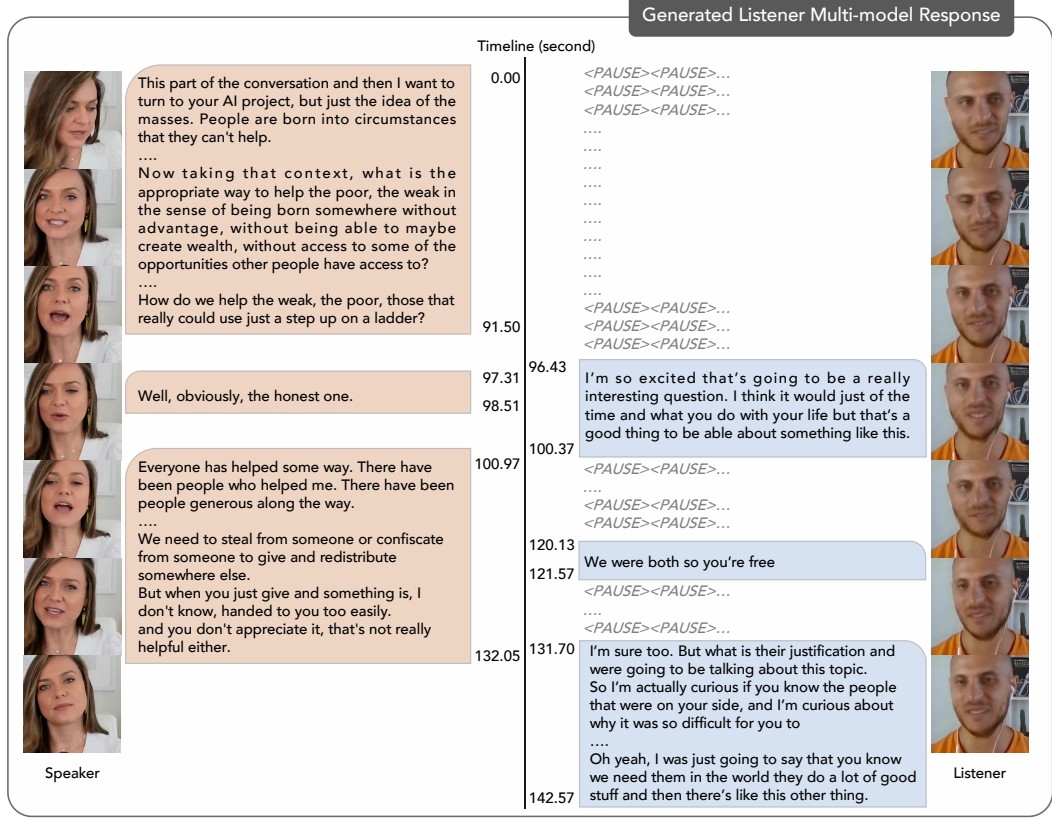

Figure 5: **Qualitative Results.** Given the speaker's audio and video streams and corresponding utterances (left), OmniResponse autoregressively generates synchronized visual, audio, and textual response streams (right). For clarity, [LASTING] tokens are removed from the generated dialogue.

conventional pipeline that integrates Automatic Speech Recognition (ASR), dialogue generation, TTS, and talking-head components waits for a predefined silence threshold before producing an offline multimodal response, thus diminishing conversational behaviors such as interruptions, backchannels, questions, and immediate feedback. In contrast, OmniResponse maintains the continuous flow of dyadic conversation by continuously modeling and generating synchronized time series streams of textual, visual, and audio outputs.

## 5.3 Ablation Studies

**Effectiveness of Chrono-Text Markup.** We construct baselines removing the proposed Chrono-Text Markup from our OmniResponse. In the baselines, each predicted word is assigned a timestamp indicating when it emerges; if this timestamp falls within a temporal window around the current time, the word is retained and appended to the spoken output; otherwise, it is discarded. As shown in the last rows of Table 3, incorporating Chrono-Text Markup significantly improves audio-visual synchronization, reducing the LSE-D score from 11.51 to 9.56. In addition, it enhances the semantic alignment of speech with conversational context, increasing METEOR from 0.122 to 0.141 and BERTScore$_{F1}$ from 0.766 to 0.806. Improvements in FD and UTMOSv2 further indicate that Chrono-Text Markup boosts the quality of the generated audio and facial responses. These results demonstrate the effectiveness of Chrono-Text Markup in generating high-quality multimodal responses.

**Effectiveness of TempoVoice.** To study the effect of our TempoVoice, we remove it from our framework and instead directly feed the hidden states, which are trimmed or padded to match the target audio length, into a multi-layer perceptron to predict audio token logits. As shown in Table 3, removing TempoVoice degrades audio–visual synchronization and reduces the quality of generated audio responses, where UTMOSv2 drops from 1.41 to 1.23, and LSE-D increases from 9.56 to 11.91.

Table 3: Ablation study on the effects of the proposed Chrono-Text Markup and TempoVoice.

| Chrono-Text Markup | Tempo Voice | METEOR | BERTScore$_{F1}$ | LSE-D | UTMOSv2 | FD |
|:---:|:---:|:---:|:---:|:---:|:---:|:---:|
| ✗ | ✗ | 0.090 | 0.755 | 13.64 | 1.21 | 596.27 |
| ✓ | ✗ | 0.128 | 0.778 | 11.91 | 1.23 | 19.58 |
| ✗ | ✓ | 0.122 | 0.766 | 11.51 | 1.39 | 23.42 |
| ✓ | ✓ | **0.141** | **0.806** | **9.56** | **1.41** | **15.46** |

Table 4: User study (A/B preference; higher is better). Each cell shows the percentage of participants preferring *Ours*.

| Criteria | Ours vs. LSTM | Ours vs. Audio–Visual LLM |
|:---|:---:|:---:|
| Speech Content Appropriateness | 75.5% | 81.6% |
| Audio Speech Quality | 77.6% | 85.7% |
| Visual Quality | 67.3% | 93.4% |
| Audio–Visual Synchronization | 91.8% | 95.9% |

These results highlight the importance of TempoVoice in temporally aligning audio with the other modalities and enhancing the quality of the generated audio.

## 5.4 User Study

We conducted a user study with 49 participants (28 male, 21 female). Each subject viewed 16 randomly ordered clips and rated speech content appropriateness, audio speech quality, visual quality, and audiovisual synchronization. All participants were proficient in English (53.1% reported advanced proficiency or daily-communication ability). Educational attainment was high: 95.9% held at least an undergraduate degree, 44.9% held a master's degree, and 18.4% held a Ph.D. Ages were distributed as follows: 14.3% under 25, 34.7% aged 26–35, 24.5% aged 36–45, and 26.5% aged 46–55. In direct A/B preferences, "Ours" achieved a minimum preference of 67.3% (speech content appropriateness vs. LSTM) and a maximum of 95.9% (audiovisual synchronization vs. Audio–Visual LLM).

## 6 Conclusion and Discussion

We have presented OmniResponse, an online multimodal generation model that produces verbal and nonverbal listener responses to a speaker's multimodal behaviors. OmniResponse integrates techniques for processing multimodal inputs, synchronizing across modalities, and aligning responses with the speaker's content. To enable evaluation of this task, Online Multimodal Conversational Response Generation in Dyadic Interactions, we introduce ResponseNet, a dataset containing parallel recordings of speaker and listener streams. Our model and dataset lay the foundation for future research in this emerging field. Experimental results demonstrate that OmniResponse significantly increases speech semantic content, audio-visual synchronisation, audio and visual quality.

**Limitations.** While our approach performed well on the evaluated datasets, the remaining challenges include the proposed approach (e.g., its results) may largely depend on the quality and diversity of training data, replying on accurate speaker–listener segmentation and can be negatively affected in noisy or overlapping conversations. Additionally, generating well-aligned multi-modal responses remains difficult in fast-changing or emotionally rich interactions, while our paper lacks fairness analysis. Future work will focus on improving these aspects.

**Risks and Potential Misuse.** This system is developed for multi-modal conversational AI, but certain risks should be acknowledged. For instance, realistic synthetic contents could be misused [59] for impersonation or misleading information. During real-time human-user interactions, users may also develop misunderstandings or excessive reliance on the system without proper contents control. To avoid these risks, we recommend clear labeling of the generated contents, appropriate usage monitoring, and the inclusion of protective measures [20, 75] (e.g., Deepfake Detection [53, 29]) against potential misuse.

**Acknowledgments.** This work is supported by the KAUST Center of Excellence for Generative AI under award number 5940. The computational resources are provided by IBEX, which is managed by the KAUST Supercomputing Core Laboratory.

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
