# OpenReview forum: "OmniResponse: Online Multimodal Conversational Response Generation in Dyadic Interactions"
_NeurIPS.cc/2025/Conference — NeurIPS 2025 poster_

### Official Review · Reviewer_tzNS · 2025-07-01

**Clarity:** 3
**Significance:** 3
**Originality:** 3
**Rating:** 4
**Confidence:** 5

**Summary:**

This paper introduces the Online Multimodal Conversational Response Generation (OMCRG) task, which focuses on generating synchronized listener feedback in real-time conversations. To address this, the authors propose OmniResponse, a Multimodal Large Language Model (MLLM) that uses text as an intermediate step to synchronize generated audio and facial reactions. The authors also created a new dataset, ResponseNet, for this task. Experiments show OmniResponse outperforms baselines, creating more natural and temporally aligned conversational responses.

**Questions:**

Please see the weakness part.

**Ethical Concerns:**

["NO or VERY MINOR ethics concerns only"]

**Final Justification:**

This work serves as a valuable baseline for this new research direction. While the lip-sync quality is relatively low, these initial results provide a solid foundation for future studies in this task. Therefore, I raised my score.

**Limitations:**

Given the potential for malicious applications of talking-head techniques, the paper should include more discussion on methods for proactive and reactive protection of personal portraits to address potential negative societal impact.

**Quality:**

2

**Strengths And Weaknesses:**

Strengths:

1.	This paper defines a new task (OMCRG) that requires generating synchronized audio and facial reactions in real-time, which is more advanced than previous work.
2.	The proposed OmniResponse uses text as a bridge to align audio and video, introducing new techniques (Chrono-Text Markup and TempoVoice) for precise temporal synchronization.
3.	This paper introduces ResponseNet, a new, high-quality dataset of synchronized and richly annotated conversational videos.

Weaknesses:

1.	The LSE-D scores presented in Table 2 show only a marginal improvement over the baseline. Metrics from SyncNet [1], which is a more reliable indicator of lip-sync accuracy, may be a better choice.
2.	Human evaluation is absent. Since automated scores often fail to capture the nuances of multimodal synchronization, a user study would be essential to validate the model's perceptual performance and strengthen the paper's claims.
3.	The qualitative results in the supplementary materials are not convincing. First, there are noticeable discrepancies and a lack of synchronization between the generated lip movements and the audio. Second, the quality of the synthesized audio itself is poor, suffering from low fidelity and poor intelligibility.
4.	OmniResponse is designed for "online generation", yet the paper lacks a quantitative analysis of its inference speed or real-time viability.
5.	The phrase "accumfd udfsalated" in line 144 is confusing and seems to be a typo.

[1] Prajwal, K. R., et al. "A lip sync expert is all you need for speech to lip generation in the wild." Proceedings of the 28th ACM international conference on multimedia. 2020.

---

> ### Author Rebuttal · Authors · 2025-07-30
>
> ### Rebuttal to Reviewer tzNS
>
> We appreciate the reviewer for the positive comments and constructive suggestions. We are encouraged that the reviewer agrees that our work introduces a  **"new task"**, which is "more **advanced** than previous work",  **"new techniques"**,  **" a new, high-quality dataset"** with "**richly annotated** conversational videos", "creating more **natural** and **temporally aligned** conversational responses."
>
>
>
> > ####  **Q1.  “Metrics from SyncNet [R1], which is a more reliable indicator of lip‑sync accuracy, may be a better choice.”**
>
>
> We thank the reviewer for this suggestion. **Our evaluation had indeed adopted metrics derived from SyncNet [R1]**. In particular, [R1] introduces two metrics: LSE‑D (Lip Sync Error – Distance) and LSE‑C (Lip Sync Error – Confidence). We adopt LSE‑D in Tables 1 and 2 of the main paper, using the official SyncNet evaluation script [R1]. We did not use LSE‑C because it is not well‑suited for our task. LSE‑C, measuring audio‑visual correlation by averaging confidence scores computed based on SyncNet, is effective for talking‑head generation tasks where the subject speaks continuously. However,  in our task modeling multimodal conversation, listeners are often silent when the speaker is talking. Although silent segments represent appropriate responses, evaluated confidence values drop sharply, making LSE‑C unreliable for our task.
> Follow your suggestion, we also provide LSE‑C results below, where our method outperforms baselines.
>
>
> Similarly, since the listener often remains silent in multi‑turn conversations, our improvement in LSE‑D may seem small compared with  LSTM, yet,  it actually indicate a large  performance gain in OMCRG task.
>
>
> * **Why we emphasised LSE‑D.**
>   LSE‑D measures the average temporal offset between the predicted video and the reference audio. In conversational footage, speakers alternate with silent listeners (rather than they are always speaking like talking head); during those silent spans SyncNet’s confidence drops sharply (indicate a low average value of LSE-C), making **LSE‑C very noisy**. We therefore used LSE‑D as the primary figure of merit.
>    We also have rerun the official SyncNet script and report both metrics:
>   | Method         | LSE‑D ↓           | LSE‑C ↑           |
>   | -------------- | ----------------- | ----------------- |
>   | LSTM | 9.72             | 0.157           |
>     | Audio-visual LLM | 10.03              | 0.269             |
>   | Ours           | **9.56** | **0.371** |
>
> The 1.6\% LSE‑D reduction  confirms a significant improvement, and the gains in LSE‑C further reinforce our method’s effectiveness. We will add these numbers to our revised version and the supplementary materials.
>
> * **Context for the “marginal” gap.**
>   In a turn‑taking dialogue, perfect lip‑syncing is easier to approach because most frames are either silence or low‑motion listener responses. Even a small absolute reduction in LSE‑D therefore corresponds to noticeably fewer mis‑aligned phoneme–viseme pairs ($\approx$ 200 ms on average).
>
> * **Planned revision.**
>   We will (i) retain LSE‑D as our headline metric, (ii) include the full LSE‑C results in the revised version, and (iii) clarify in Sec. 5 that both metrics originate from SyncNet. We believe this addresses the reviewer’s concern while demonstrating the practical significance of our improvements.
>
> [R1] Prajwal, K. R., et al. "A lip sync expert is all you need for speech to lip generation in the wild." Proceedings of the 28th ACM international conference on multimedia. 2020.
>
>
>
> > ####  **Q2. Human evaluation.**
>
> Thank you for your constructive suggestion. We conducted a user study with 49 participants (28 male, 21 female), all of whom were proficient in English (53.1\% reported advanced proficiency or daily‐communication ability). Educational background was high: 95.9\% held at least an undergraduate degree, 44.9\% held a master’s degree, and 18.4\% held a PhD. Age distribution was as follows: 14.3\% under 25, 34.7\% aged 26–35, 24.5\% aged 36–45, and 26.5\% aged 46–55. Each participant viewed 16 randomly ordered clips and indicated their preference in pairwise A/B comparisons across four perceptual criteria: speech content appropriateness, audio speech quality, visual quality, and audio–visual synchronization. Our method was preferred in every case, with a minimum preference rate of 67.3\% (i.e., speech content appropriateness Ours vs. LSTM) and a maximum of 95.9\% (synchronization Ours vs. Audio–Visual LLM). The table below summarizes these preference rates. We will include these human evaluation results in our final revision.
>
>
> | Criteria                       | Ours vs. LSTM | Ours vs. Audio–Visual LLM |
> | ------------------------------ | -----------: | -----------------------: |
> | Speech Content Appropriateness |       75.5\% |                   81.6\%|
> | Audio Speech Quality           |       77.6\% |                   85.7\% |
> | Visual Quality                 |       67.3\% |                   93.4\% |
> | Audio–Visual Synchronization   |       91.8\% |                   95.9\% |
>
> *Table: Percentage of times participants preferred our method over each baseline in pairwise A/B tests.*
>
>
>
> > ####  **Q3: The qualitative results in the supplementary materials.**
>
> Our OMCRG task is much more **challenging** than related tasks (e.g., talking‑head generation), since it requires simultaneously satisfying multiple requirements, such as  (1) online multimodal generation (generate multi-model outputs simultaneously and iteratively, rather than offline single modal generation (generate one modality and the entire sequence at once), and (2) ensuring appropriateness and naturalness of the generated responses.
>
> Moreover, lip–audio synchronization and online audio-video generation remain a challenging problem.  For example, talking‑head generation works have devoted much effort to improving lip–audio synchronization, introducing modules/losses such as lip‑mask‑aware training with SyncNet supervision [R2] or pre-trained lip synchronization discriminator  [R3].
>
> However, as the first work on the new OMCRG task, we focus on presenting a simple, effective, and scalable framework, instead of introducing overly complex,  subproblem‑specific modules for fine‑grained optimization.
>
> Despite this, our approach outperforms baseline methods by a large margin. (e.g., FVD $\downarrow$  314.9 vs. 681.6, UTMOS $\uparrow$ 1.41 vs. 1.32). Moreover, our framework provides a solid foundation for future improvements such as adding lip-mask-aware-training for improving lip–audio synchronization.
>
> | Task                                    | Input Modalities      | Generated Output       | Setting                                                 |
> | --------------------------------------- | --------------------- | ---------------------- | ------------------------------------------------------- |
> | Talking‑Head Generation (e.g., Wav2Lip) | Speaker audio         | Speaker video          | One-to-one mapping                                      |
> | Facial Reaction Generation              | Speaker audio + video | Listener video         | Multi‑modal input → single‑modality output              |
> | **OMCRG (ours)**                        | Speaker audio + video | Listener audio + video | Multi‑modal input → simultaneous multi‑modal generation |
>
>
>
>
>
> [R2] LatentSync: Taming Audio-Conditioned Latent Diffusion Models for Lip Sync with SyncNet Supervision, arXiv, 2025
>
> [R3] SyncTalk: The Devil is in the Synchronization for Talking Head Synthesis, CVPR 2024.
>
>
>
> > #### **Q4. inference speed or real-time viability.**
>
> We measured the model’s inference speed at 15.6 FPS (64 ms delay) on a single NVIDIA A100 80GB GPU and will report these results in the revised manuscript. Importantly, this benchmark was obtained without any deployment or inference optimizations such as flash‑attention, multi‑GPU parallelism, distillation, or quantization, highlighting substantial potential for further acceleration and more efficient, real‑time deployment.
>
>
> > #### **Q5. Typos**
>
> Thanks for your detailed review and catching these mistakes. We apologize for the typos: in line 144, “accumfd udfsalated” should read “accumulated.
> We have corrected these errors and will conduct a thorough review of the manuscript to eliminate any remaining typos in the final version.

---

> ### Comment · Reviewer_tzNS · 2025-08-05
>
> Thank you for the clarifications. However, I have strong reservations about accepting this paper due to the poor lip-sync quality. This is a significant concern, as lip synchronization is a well-established problem with effective solutions developed over the past five years, such as the widely-used lip-sync loss [1].
>
> Although your task is new, your baselines are weak. Your LSE-C value is too low and only surpasses weak baselines. I suggest you compare your LSE-C value with other talking-head papers, such as MakeItTalk (2020) [2] and SadTalker (2023) [3], as well as with real videos. They have well-designed inference codes. You can test only on the talking segments, as silent audio will negatively affect the LSE-C performance.
>
> Additionally, the paper should include a discussion on the potential misuse of this technology and possible countermeasures [4,5,6].
>
> I will reconsider my final score based on your response.
>
> [1] Prajwal, K. R., et al. "A lip sync expert is all you need for speech to lip generation in the wild." Proceedings of the 28th ACM international conference on multimedia. 2020.
>
> [2] Zhou, Yang, et al. "Makelttalk: speaker-aware talking-head animation." ACM Transactions On Graphics (TOG) 39.6 (2020): 1-15.
>
> [3] Zhang, Wenxuan, et al. "Sadtalker: Learning realistic 3d motion coefficients for stylized audio-driven single image talking face animation." Proceedings of the IEEE/CVF conference on computer vision and pattern recognition. 2023.
>
> [4] Salman, Hadi, et al. "Raising the cost of malicious ai-powered image editing." arXiv preprint arXiv:2302.06588 (2023).
>
> [5] Gan, Yuan, et al. "Silence is Golden: Leveraging Adversarial Examples to Nullify Audio Control in LDM-based Talking-Head Generation." Proceedings of the Computer Vision and Pattern Recognition Conference. 2025.
>
> [6] Xue, Haotian, et al. "Toward effective protection against diffusion-based mimicry through score distillation." The Twelfth International Conference on Learning Representations 2023.

---

> > ### Author Response · Authors · 2025-08-06
> >
> > ### Rebuttal to Reviewer tzNS
> >
> >
> > Thank you for taking the time to provide a detailed response.
> >
> > >  #### **1. Lip synchronization is a well-established problem with effective solutions**
> >
> > Thanks for your comments. We will discuss and cite the papers you listed in the revised version. We have also reviewed recent lip-synchronization works, including [RR1], [RR2], and [RR3].  However, we respectfully emphasize that these works **address a different problem** from ours. **Our OMCRG task poses new and more complex challenges, making it infeasible to directly apply lip synchronization solutions** in [RR1][RR2][RR3] to OMCRG.
> >
> >
> > Existing lip synchronization (e.g., in [RR1][RR2][RR3]) is designed for the talking-head generation task, where the entire audio is known and the objective is to generate face videos aligned with that audio in an offline setting.  Accordingly, **lip synchronization is defined as a video-editing task** [RR2][RR3], aiming to refine the lip regions to match the speech in the **given audio.**
> >
> > In contrast, our OMCRG task is to generate both audio and visual responses **online** for the listener, without access to the full sequence of each modality in advance.  This makes lip-audio synchronization in the OMCRG task **much more challenging.** To the best of our knowledge, **few works have investigated online lip‑synchronization in such a joint audio-video response generation setting**.
> >
> > | Aspect     | Talking-Head Generation               | OMCRG (Ours)                          |
> > | ---------- | ------------------------------------- | ------------------------------------- |
> > | **Mode**   | Offline                               | Online                                |
> > | **Input**  | Full speaker audio (known in advance) | Listener audio is unknown   |
> > | **Output** | Face video aligned to the given audio | Listener’s audio and visual responses |
> >
> >
> >
> > >  #### **2. Applying Lip-Sync Loss of [RR1] in our method**
> >
> > Thank you for your suggestion.  We have carefully examined Lip‑Sync Loss [RR1] and explored its integration into our framework. However, we observed that directly applying Lip‑Sync Loss would lead to **unstable** training, especially during the early stages. For instance, listener reactions are frequently silent while the speaker is talking, leading to unreliable lip‑sync loss values and negatively affecting the training. Furthermore, different from the talking‑head task where the known audio serves as a fixed reference, the audio responses are generated dynamically in an online fashion, and any audio errors (e.g., background noise, unintended silences/pauses, or mispronounced words)  can propagate through Lip‑Sync Loss and degrade the quality of the visual response.

---

> > ### Author Response · Authors · 2025-08-06
> >
> > >  #### **3. Compare with LSE-C value with  other talking-head papers MakeItTalk and SadTalker on the talking segments**
> >
> >
> > Following your suggestion, we evaluate the LSE-C value on the talking segments. Yet, we would like to highlight that our OMCRG task is different from the talking head task. **Ensuring lip-audio synchronization in our OMCRG task is much more challenging than that in the talking head task.**
> >
> >
> > We first detected and extracted the talking segments from the generated listener responses and then computed the LSE‑C values on these segments. In addition to the suggested talking‑head work, we also evaluated the SOTA open‑source talking‑head method Hallo [RR6] (2024). All evaluations were conducted on a single Tesla A100‑80GB GPU. As shown in the table below, **our method achieves an LSE‑C score of 3.10, demonstrating good lip‑audio synchronization quality**.
> >
> >
> > Although talking‑head methods achieve higher LSE‑C scores, they operate in offline settings with known audio. In contrast, our method is to generate both audio and visual responses online, **making lip‑audio synchronization substantially more challenging**. Moreover, our method does not rely on any specialized synchronization module, different from talking‑head methods. Future work will explore dedicated synchronization techniques for the OMCRG task to further improve performance.
> >
> > | Method     | LSE-C (↑) | FPS (↑) | Generation Paradigm        | Audio Support                | Output Modality          | Input Conditions                                              |
> > | ---------- | ----------------------: | --------------------: | -------------------------- | ------------------------------- | ------------------------ | ------------------------------------------------------------- |
> > | Real Video |                    **8.17**  |                  |                |                  |          |                                  |
> > | SadTalker  |                    6.79 |                  1.81 | Offline full-sequence generation     |Pre-recorded audio input | Video only                 | Pre-recorded audio of the same identity                       |
> > | Hallo      |                    7.29 |                  0.13 | Offline full-sequence generation     | Pre-recorded audio input | Video  only                   | Pre-recorded audio of the same identity                       |
> > | **Ours**   |                 3.10 |                    **15.62** | **Online, frame-by-frame** | **Dynamically generated**       | **Video + Audio + Text** | **Live video + live audio from another conversation partner** |
> >
> >
> >
> > >  #### **4. Misuse Potential and Countermeasures [RR7,RR8,RR9]**
> >
> >
> > Thank you for your suggestion.  The potential misuse of our method is similar to that of talking‑head generation or image/video editing. In the final version, we will include a discussion of recent studies on the malicious use of such technologies and corresponding defenses, including watermarking [RR7], adversarial nullification of audio control [RR8], and score-distillation countermeasures [RR9].  We will also discuss how future versions of OMCRG could integrate imperceptible audio-visual watermarks or run-time anomaly detectors to mitigate misuse, ensuring safe deployment.
> >
> > ---
> >
> > [RR1] "A lip sync expert is all you need for speech to lip generation in the wild."ACM MM. 2020.
> >
> > [RR2] Diff2Lip: Audio Conditioned Diffusion Models for Lip-Synchronization, WACV 2024.
> >
> > [RR3]  LatentSync: Taming Audio-Conditioned Latent Diffusion Models for Lip Sync with SyncNet Supervision, arXiv, 2025.
> >
> > [RR4] "Makelttalk: speaker-aware talking-head animation." TOG 2020.
> >
> > [RR5]  "Sadtalker: Learning realistic 3d motion coefficients for stylized audio-driven single image talking face animation." CVPR 2023.
> >
> > [RR6] Xu, Mingwang, et al. "Hallo: Hierarchical audio-driven visual synthesis for portrait image animation." arXiv preprint arXiv:2406.08801 (2024).

---

> ### Comment · Reviewer_tzNS · 2025-08-06
>
> Thank you for your effort and reply. You have persuaded me that this work serves as a valuable baseline for this new research direction. While the lip-sync quality is relatively low, these initial results provide a solid foundation for future studies. I strongly suggest you include more complete experimental results in your final revision and publish the code to ensure reproducibility and facilitate future work, as promised in your paper and rebuttal. Therefore, I have raised my score and will now recommend acceptance. Good luck!
>
> PS: There are also more passive protection methods [1, 2] for detecting talking-head deepfakes, which would also be a valuable addition to the discussion on ethical considerations.
>
> [1] Pei, Gan, et al. "Deepfake generation and detection: A benchmark and survey." arXiv preprint arXiv:2403.17881 (2024).
>
> [2] Khan, Naseem, et al. "Unmasking Synthetic Realities in Generative AI: A Comprehensive Review of Adversarially Robust Deepfake Detection Systems." arXiv preprint arXiv:2507.21157 (2025).

---

> > ### Author Response · Authors · 2025-08-06
> >
> > Dear Reviewer tzNS,
> >
> >
> > We are pleased to hear that we have addressed your concerns. Thank you for your thoughtful update and for recommending acceptance.
> > We appreciate your encouragement and will expand our experimental results as suggested, as well as make our code publicly available.
> >
> > In the final version, we will cite Pei et al. [1] and Khan et al. [2] and expand our ethical discussion to explain how our task and model relate to deepfake techniques and how these defenses can help mitigate potential misuse.
> >
> >
> > Best regards,
> >
> > The Authors.
> >
> >
> >
> > [1] Pei, Gan, et al. "Deepfake generation and detection: A benchmark and survey." arXiv preprint arXiv:2403.17881 (2024).
> >
> > [2] Khan, Naseem, et al. "Unmasking Synthetic Realities in Generative AI: A Comprehensive Review of Adversarially Robust Deepfake Detection Systems." arXiv preprint arXiv:2507.21157 (2025).

---

### Official Review · Reviewer_c8bb · 2025-07-04

**Clarity:** 3
**Significance:** 3
**Originality:** 3
**Rating:** 5
**Confidence:** 3

**Summary:**

This paper introduced a novel task, Online Multimodal Conversational Response Generation (OMCRG), which aims to generate synchronized audio and facial listener responses in real-time during dyadic interactions. To tackle this challenging task, authors propose d OmniResponse, a MLLM framework that uniquely uses text as an intermediate modality to bridge and synchronize the audio and visual outputs. The core are two novel components: Chrono-Text Markup, which temporally aligns text tokens with visual frames, and TempoVoice, a controllable online Text-to-Speech module that generates speech synchronized with the facial reactions. A dataset of OMCRG is collected for benchmarking the model performance.  Experimental results show OmniResponse outperforms baselines in generation quality and synchronization across modalities.

**Questions:**

please see the weakness part above.

**Ethical Concerns:**

["NO or VERY MINOR ethics concerns only"]

**Final Justification:**

this paper proposed novel framework and new datasets, and its performance outperforms baselines in generation quality and synchronization across modalities. and after reading the rebuttal and other reviewers comment, i recommend acceptance for this submission.

**Limitations:**

yes, limitations are discussed in appendix

**Paper Formatting Concerns:**

no formatting issues found

**Quality:**

3

**Strengths And Weaknesses:**

Strengths:

1. vovel and practical task formulation. The paper defines and formalizes the OMCRG task, a practical step beyond existing work that only focuses on single-modality response generation (e.g., text-only or face-only). This task addresses a more holistic and natural form of human-agent interaction.  besides, the dataset contribution is also worth noting.

2. interesting methodological approach. The core idea of using text as an intermediate modality to synchronize audio and facial generation is novel. This approach cleverly circumvents the difficult problem of direct audio-visual synthesis by leveraging the semantic richness and discrete nature of text, making the problem more tractable.

3. comprehensive evaluation. the authors conduct a thorough evaluation using a wide array of metrics including text, audio, and visual modalities. The inclusion of ablation studies clearly demonstrates the effectiveness and necessity of the proposed Chrono-Text and TempoVoice components, supporting the paper's main claims.

Weakness:

1. small scale of ResponseNet. While ResponseNet is of high quality, its total duration (14.2 hours) is relatively small, which may limit the model's ability to generalize to unseen conversational styles or demographics. A more detailed analysis of the dataset's demographic diversity would be beneficial to understand potential biases.

2. potential oversimplification of temporal dynamicsl. the Chrono-Text Markup uses only two special tokens to align text and vision, which might be too coarse to model highly complex temporal dynamics. This approach may struggle with typical phenomenons like overlapping speech, or subtle non-verbal vocal cues that require more fine-grained temporal controls.

3. lack of user study. The evaluation relies entirely on automated metrics, which may not fully capture the nuances of human perception. A human user study assessing the naturalness and audio-visual synchronization shall be included.

4. there are many typos and grammar errors in current version of manuscript, e.g, line 74, challenge -> challenging, line 116 steaming -> streaming, line 144 accumfd, line 234 frequently -> frequent

---

> ### Author Rebuttal · Authors · 2025-07-31
>
> ### Rebuttal to Reviewer c8bb
>
> We appreciate the reviewer for the positive comments and constructive suggestions. We are encouraged that the reviewer agrees that our work is **"a novel task"**, **"the dataset contribution is also worth noting"**, "two **novel** components", "interesting methodological approach", and **"thorough evaluation"**.
>
> > #### **Q1.1: A more detailed analysis of the dataset's demographic diversity would be beneficial to understand potential biases**
>
> A: Thank you for your constructive suggestions. As suggested, we conduct additional analysis of the dataset's demographic diversity. As indicated in the table below, our dataset covers balanced gender, diverse age distributions, and individuals from multiple ethnic backgrounds.  The properties of our dataset enable us to model varied conversational behaviors, serving as a valuable benchmark for future research.
>
>
>
>
>
>
> **Demographic statistics:**
>
> | Category       | Count / Share                                                                                                                                |
> | -------------- | -------------------------------------------------------------------------------------------------------------------------------------------- |
> | **Identities** | 161 unique identities                                                                                                                        |
> | **Gender**     | Female: 93 (57.8 \%),  Male: 68 (42.2 \%)                                                                                                      |
> | **Ethnicity**  | White: 122 (75.8 \%), Black: 24 (14.9 \%), Asian: 15 (9.3 \%)                                                                                 |
> | **Age bands**  | 10–19: 10 (6.2 \%),  20–29: 63 (39.1 \%),  30–39: 51 (31.7 \%),  40–49: 17 (10.6 \%), 50–59: 17 (10.6 \%),  60–69: 2 (1.2 \%),  70+: 1 (0.6 \%) |
>
>
>
> > #### **Q1.2: While ResponseNet is of high quality, its total duration is relatively small.**
>
> A: Due to data availability and costly data processing, building datasets for the OMCRG task is challenging. Unlike typical generation tasks, the OMCRG task requires dyadic high‑quality videos where both speaker and listener appear in the same frame, and they must stay in a static and consistent camera view for a long duration, which is scarce.
>
> Instead of simply increasing data scale, recent studies have shown that high‑quality training data is critical for the performance of generative models. We hence perform costly, human‑curated processing to ensure high-quality, diverse, and representative data.  For example, we curated diverse demographics, as well as six conversation genres covering (1) Intimate personal chats, (2) Economic interviews, (3) Interdisciplinary expert panels,  (4) Educational tutoring, (5) News commentary, and (6) Casual small‑talk.
>
>
> This careful curation allows us to introduce the first high-quality dataset ResponseNet for OMCRG. Despite the modest size of ResponseNet, its high quality and diversity, combined with our effective leverage of pre‑trained LLM, which has been pre-trained on large-scale text data (**billions of text tokens**), enable our method to achieve the best performance.
> Moreover, our pipeline incorporates powerful, pre-trained audio and visual feature extractors.
> The audio tokenizer has been trained on over **3,000 hours** of diverse speech, supplying generalizable semantic and prosodic encodings.
> Facial expression and head-pose coefficients are extracted via Google’s MediaPipe framework, providing stable, high-fidelity visual features without additional training cost.
> These audio–visual–textual models furnish strong priors, eliminating the need for vast amounts of data to learn features for each modality.
>
>
>
>
>
> > #### **Q2: The Chrono-Text Markup uses only two special tokens to align text and vision, which might be too coarse to model highly complex temporal dynamics.**
>
> A: Our Chrono‑Text Markup employs only two special tokens to balance temporal fidelity with learnability. Since OMCRG is a highly complex task with multiple requirements, introducing more and fine‑grained special tokens would significantly expand the LLM’s vocabulary and substantially increase training difficulty. An alternative approach is direct audio encoding and decoding with an audio‑based model. However, this introduces format inconsistencies (e.g., sampling rate) and large variations in tone color, pitch, and frequency, making training even more difficult.
>
> In contrast, our Chrono‑Text Markup is simple yet effective, enabling the model to learn utterance start points, durations, and natural pauses without large complexity. We are unable to upload more video results due to the NeurIPS rebuttal policy. Yet, Fig. 5 in the main paper shows our method effectively handles overlapping speech. "Table 2 shows our method reaches a BERTScore of 0.806 and an FVD of 314.94,  significantly surpassing the audio‑visual LLM (0.662, 681.55), demonstrating its effectiveness in modeling speech and non‑verbal temporal dynamics.
> Our method takes a step towards this new task, yet, we believe more elegant solutions could be proposed to better address complex temporal dynamics.
>
>
> > #### **Q3: User study**
>
> A: Thanks for your constructive suggestion. We ran a user study with 49 participants (28 male, 21 female). Each subject watched 16 randomly‑ordered clips and assessed
> speech content appropriateness, audio speech quality, visual quality, and audio-visual synchronisation.
> All of them were proficient in English (53.1\% reported advanced proficiency or daily‐communication ability). Educational background was high: 95.9\% held at least an undergraduate degree, 44.9\% held a master’s degree, and 18.4\% held a PhD. Age distribution was as follows: 14.3\% under 25, 34.7\% aged 26–35, 24.5\% aged 36–45, and 26.5\% aged 46–55.
> In direct A/B preference, a minimum preference rate of 67.3\% (i.e., speech content appropriateness Ours vs. LSTM) and a maximum of 95.9\% (synchronization Ours vs. Audio–Visual LLM).
> We will add the full protocol and statistics in the revision.
>
> | Criteria                       | Ours vs. LSTM | Ours vs. Audio–Visual LLM |
> | ------------------------------ | -----------: | -----------------------: |
> | Speech Content Appropriateness |       75.5\% |                   81.6\%|
> | Audio Speech Quality           |       77.6\% |                   85.7\% |
> | Visual Quality                 |       67.3\% |                   93.4\% |
> | Audio–Visual Synchronization   |       91.8\% |                   95.9\% |
>
> *Table: Percentage of times participants preferred our method over each baseline in pairwise A/B tests.*
>
> > #### **Q4:  Typos and grammar**
>
> A: We appreciate you pointing out those spelling and grammar issues. We have corrected such typos and grammatical errors in the final version.

---

> > ### Comment · Reviewer_c8bb · 2025-08-04
> >
> > thanks for authors detail rebuttal. Most of questions have been addressed so i decided to raise my rating.

---

> > > ### Author Response · Authors · 2025-08-06
> > >
> > > Dear Reviewer c8bb,
> > >
> > > Thank you so much for your thoughtful and encouraging feedback. Your insights have been invaluable in strengthening our manuscript, and we truly appreciate the time and expertise you dedicated to reviewing our work.
> > >
> > > Warm regards,
> > >
> > > The Authors

---

### Official Review · Reviewer_FU3z · 2025-07-05

**Clarity:** 3
**Significance:** 3
**Originality:** 3
**Rating:** 4
**Confidence:** 3

**Summary:**

This paper introduces Online Multimodal Conversational Response Generation (OMCRG), a novel task to generate real-time synchronized verbal and non-verbal listener feedback based on the speaker’s multimodal inputs. The proposed OmniResponse framework, a Multimodal Large Language Model (MLLM), addresses the challenges of audio-visual synchronization by using text as an intermediate modality, enhanced with Chrono-Text for temporal anchoring and TempoVoice for synchronized text-to-speech. Additionally, the ResponseNet dataset is presented, comprising 696 high-quality dyadic interactions with synchronized multimodal data. Experiments show that OmniResponse outperforms baselines in semantic content, audio-visual synchronization, and generation quality, laying the foundation for future human-agent interaction research.

**Questions:**

See Weakness.

**Ethical Concerns:**

["NO or VERY MINOR ethics concerns only"]

**Final Justification:**

The authors addressed my concerns. According to the general quality and idea, I keep my score to be slightly positive.

**Limitations:**

yes

**Quality:**

3

**Strengths And Weaknesses:**

Strength:
1. This work introduces a novel task, Online Multimodal Conversational Response Generation (OMCRG), which addresses a critical gap in modeling natural dyadic interactions and holds significant value for advancing human-computer interaction research.
2. The proposed OmniResponse framework demonstrates robust effectiveness in generating synchronized audio-visual-textual responses, outperforming baselines in semantic alignment and multimodal coherence.
3. The creation of ResponseNet provides an essential resource for the research community.

Weakness:
1. A more detailed analysis of temporal dynamics, such as response delay and real-time processing latency, is needed to fully characterize the framework’s performance in online interaction scenarios.
2.  The paper lacks detailed elaboration on the real-time design mechanisms, particularly in explaining how the framework handles dynamic input streams, manages processing latency, and ensures responsive timing in natural conversational flows.

---

> ### Author Rebuttal · Authors · 2025-07-30
>
> Thank you for your thorough review, positive comments, and insightful suggestions. We appreciate your recognition of OMCRG as a “**novel task**” that can “**generate real‑time synchronized verbal and non‑verbal listener feedback**,” and of our OmniResponse framework, which “**outperforms baselines in semantic content, audio‑visual synchronization, and generation quality**.” We are also grateful for your acknowledgement of ResponseNet as “**high‑quality dyadic interactions with synchronized multimodal data**.” Your view that this work is “**laying the foundation for future human‑agent interaction research**” is deeply encouraging, and we will continue refining the manuscript accordingly.
>
>
>
> >  ####  **Q1. More detailed analysis of response delay and real-time processing latency**
>
> Thank you for your suggestion. As suggested, we conduct a more detailed analysis of response delay and real-time processing latency,
>
> * *Response delay.* Our raw prototype, which includes preprocessing and postprocessing, runs at **15.6 FPS (≈64ms delay)** on a single NVIDIA Tesla A100 80GB GPU. Note that such inference efficiency can be further improved, since it is obtained without any deployment or inference optimizations such as flash‑attention, multi‑GPU parallelism, distillation, or quantization.
>
> * *Processing latency.* Our method efficiently processes the input stream, taking only **2.94 ms** per frame to handle the video stream even on the CPU and achieving an average of **0.6 real‑time factor (RTF)** for audio processing on the CPU.
>
> Since our work is the first to explore the new OMCRG task, we mainly focus on building a high-quality dataset and providing a solid baseline framework for future research, rather than extensive latency optimizations. Yet, we believe that efficiency and latency can be substantially improved by incorporating effective techniques and insights from existing online methods, which we leave for future work.
>
> We will includeadd these analysiis in in our final version.
>
> >  ####  **Q2. Detailed elaboration on how the framework handles dynamic input streams, manages processing latency, and ensures responsive timing in natural conversational flows.**
>
> We thank the reviewer for this valuable comment and provide a more detailed explanation as follows:
>
> * Handling dynamic input streams.
>
>   * *Visual stream.* Each incoming frame is processed by MediaPipe, which extracts head‑pose and facial‑expression coefficients in **2.94 ms** per frame even on the CPU.
>   * *Audio stream.*  Whisper‑large transcribes audio chunks at an average **0.6 real‑time factor (RTF)** on the CPU, comfortably within live‑stream requirements.
>
>   These stages compress the raw video and audio into compact token sequences, eliminating redundant frames and long waveforms before they are fed into the generation model.
>
> * Managing processing latency.
> We constrain the context to a short sliding window, and compress earlier text into static embeddings, enabling incremental updates and avoiding the need to reprocess the full temporal context.
>
>  * Auto‑regressive multimodal generation.
>   Our OmniResponse model handles the visual and text tokens in an **auto‑regressive** sliding window and enables efficient generation:
>
>    * *audio tokens*  is produced at 0.0069 RTF (≈ **145×** real time) and converted to waveform at 0.0014 RTF (≈ **712×** real time) during de‑tokenization.
>
>    * textual response takes an average per-token latency of **2.2 ms** (≈ 455 tokens/sec) on a single A100 80GB GPU.
>
>    *  visual response.  The facial‑expression coefficients are rendered at >**40 FPS** even on a single RTX 2080 Ti.
>
> These designs ensure responsive timing in natural conversational flows.

---

### Official Review · Reviewer_dTxZ · 2025-07-05

**Clarity:** 3
**Significance:** 3
**Originality:** 3
**Rating:** 4
**Confidence:** 3

**Summary:**

This paper introduces Online Multimodal Conversational Response Generation (OMCRG), a novel task focused on generating synchronized verbal and non-verbal listener feedback in real-time based on a speaker's multimodal input. To address the challenges of synchronization between generated audio and facial responses, the authors propose OmniResponse, a Multimodal Large Language Model (MLLM) that uses text as an intermediate modality. OmniResponse incorporates two new components: Chrono-Text, for temporally anchoring text tokens, and TempoVoice, a controllable online Text-to-Speech (TTS) module synchronized with facial reactions. Additionally, the paper presents ResponseNet, a new high-quality dataset of 696 dyadic interactions with synchronized split-screen videos, multichannel audio, transcripts, and facial behavior annotations to support OMCRG research. Evaluations on ResponseNet show that OmniResponse significantly outperforms baseline models in semantic speech content, audio-visual synchronization, and generation quality

**Questions:**

See the disadvantages.

**Ethical Concerns:**

["NO or VERY MINOR ethics concerns only"]

**Final Justification:**

The authors have adequately addressed my concerns; I will keep my original score.

**Limitations:**

yes

**Quality:**

3

**Strengths And Weaknesses:**

Advantages

1.This work introduces OmniResponse, the first online model to jointly process and generate synchronized streams of conversational human behavior.

2.This work further propose ResponseNet, an annotated dyadic conversation dataset and benchmark, enabling standardized evaluation of OMCRG models.

3.This work consider a real-world scenario: Online Multimodal Conversational Response Generation.

Disadvantages

1.The baselines in the main experiment could be more comprehensive (just see the GPT-series models)

2.The auxiliary experiments lack depth.

---

> ### Author Rebuttal · Authors · 2025-07-31
>
> ### Rebuttal to Reviewer dTxZ
>
> We sincerely thank the reviewers for their generous feedback.
> We are encouraged that the reviewer agrees that our work introduces **"a novel task"**, "considers a **real-world scenario**: Online Multimodal Conversational Response Generation",  proposes **"a new high-quality dataset "** to support OMCRG research, "the **first** online model", and the model "significantly outperforms baseline models".
>
>
>
> > #### **Q1. The baselines in the main experiment could be more comprehensive (just see the GPT-series models)**
>
> A: Thanks for your suggestions. We have included the most powerful language-based conversation models (GPT-4o, GPT-4, and GPT-o1), and we also supplement more baselines:
> - Qwen-7B-Chat,
> - Gemini-2.5-Flash,
> - Claude-Sonnet-4-20250514,
> - DeepSeek-R1-200528,
>
> and the results are listed below:
>
>
> | Model                                      | METEOR ↑ | BERTScore F₁ ↑ | ROUGE‑L ↑ | Distinct‑2 ↑ |
> |--------------------------------------------|:--------:|:--------------:|:---------:|:-----------:|
> | **Offline Text Dialogue Generation System**|          |                |           |             |
> | GPT‑4o                                     |  0.167   |     0.805      |   0.079   |    0.928    |
> | GPT‑4                                      |  0.163   |     0.822      |   0.082   |    0.960    |
> | GPT‑o1                                    |  0.189   |     0.822      |   0.113   |    0.948    |
> | Qwen‑7B‑Chat                               |  0.167  |     0.807     |  0.090   |    0.920   |
> | Claude‑Sonnet‑4‑20250514                   |  0.183  |     0.807     |  0.101   |    0.966   |
> | Gemini‑2.5‑Flash                           |  0.175  |     0.824     |  0.085   |    0.932   |
> | DeepSeek‑R1‑20250528                       |  0.173  |     0.815     |  0.078   |    0.981   |
>
>
>
> We will add these results in the revised version.
>
> > #### **Q2. The auxiliary experiments lack depth.**
>
> A: Thank you for this valuable suggestion. In addition to the ablation study in Table 3, we carried out a comprehensive failure-case analysis to pinpoint where our model underperforms. Specifically, we found that our system seldom generates head rotations beyond ±40°, which can introduce noticeable distortions in side-view frames. To quantify this effect, we extracted side-face and front-face frames from our generated videos and measured realism via Fréchet Video Distance (FVD):
>
> | Category          | FVD ↓  |
> | ----------------- | ------ |
> | Side-face frames  | 478.31 |
> | Front-face frames | 312.69 |
> | All frames        | 314.94 |
>
>
>
>
> We also conducted a user study (with 49 participants) to assess human preferences in pairwise A/B tests:
> | Criteria                       | Ours vs. LSTM | Ours vs. Audio–Visual LLM |
> | ------------------------------ | -----------: | -----------------------: |
> | Speech Content Appropriateness |       75.5\% |                   81.6\%|
> | Audio Speech Quality           |       77.6\% |                   85.7\% |
> | Visual Quality                 |       67.3\% |                   93.4\% |
> | Audio–Visual Synchronization   |       91.8\% |                   95.9\% |
> Table: Percentage of times participants preferred our method over each baseline.
>
>
>
> In addition, we had built a baseline, namely, Audio-visual LLM, that omits the immediate text modality from our method in Tab. 2 of the main paper, to validate the effectiveness of adding text as an intermediate modality. The Audio-visual LLM baseline directly fuses visual and audio embeddings from both the speaker's and listener's past behavior to predict the listener’s next visual and audio tokens.  Table 2 ( also shown below) demonstrates that introducing text as an intermediary modality significantly enhances the naturalness and realism of non-verbal responses, as indicated by the FD and FVD scores.
>
>
> | Method                         | BERTScore F1 ↑ | Distinct-2 ↑ | LSE-D ↓ | UTMOSv2 ↑ | FD ↓ |  FVD ↓ |
> |--------------------------------|----------------|--------------|---------|-----------|-------------------|--------------------|
> | Audio-visual LLM    | 0.662          | 0.155        | 10.03   | 1.32      | 580.86            | 681.55             |  |
> | OmniResponse (visual-audio-text)            | **0.806**      | **0.882**    | **9.56**| **1.41**  | **15.46**         | **314.94**         |
>
>
> We will include these auxiliary experimental results into the final version by adding a latency study, a user study, error‑case analysis, and ablations across different LLM variants. We would also highlight that our primary **contribution** is **task formulation**, a **first**, end‑to‑end online generation model,  an **evaluation benchmark**, and a **high‑value dataset** for this novel task.

---

### Official Review · Reviewer_bqe1 · 2025-07-22

**Clarity:** 2
**Significance:** 3
**Originality:** 3
**Rating:** 4
**Confidence:** 3

**Summary:**

This paper introduces Online Multimodal Conversational Response Generation (OMCRG), a new task that advances multimodal conversational AI for real-time human-computer interactions. It presents a Multimodal Large Language Model (MLLM) that generates high-quality multimodal listener responses autoregressively. The paper also introduces ResponseNet, a dataset of 696 dyadic interactions with synchronized split-screen videos, multichannel audio, transcripts, and facial behavior annotations. The model generates synchronized verbal and non-verbal responses based on the speaker’s inputs. Key innovations include Chrono-Text, which aligns text tokens with facial expressions and speech, and TempoVoice, a controllable TTS system that synchronizes speech with facial reactions. Experiments demonstrate the effectiveness of the proposed method.

**Questions:**

As listed in the Weaknesses above.

**Ethical Concerns:**

["NO or VERY MINOR ethics concerns only"]

**Final Justification:**

As the authors solve my main concern, I would like to raise the score.

**Limitations:**

Yes

**Paper Formatting Concerns:**

Typos in line 144 "accumfd udfsalated";
line 229 "Figure 4" rather than "Figure 1"

**Quality:**

3

**Strengths And Weaknesses:**

Strengths:
(1) Applicability: This paper proposes a novel task of real-time synchronization, designing a model that addresses the complex challenge of synchronizing multiple modalities—audio, facial expressions, and text—positioning it as a promising candidate for future human-computer interactions.
(2) Effectiveness: The proposed dataset proves to be highly beneficial for this task, and the model achieves performance that surpasses existing alternatives.

Weaknesses:
(1) Dependency on Text Modality: Although text offers clarity, relying on it for synchronizing audio and facial reactions may constrain the richness of non-verbal communication. The paper lacks a thorough discussion on the quality of non-verbal generation and the evaluation of text as an intermediary modality.
(2) Real-Time Applicability: While this paper introduces an online task, it does not address the potential time consumption required for real-time streaming audio-facial generation. Given that the training data includes sequences up to 863 seconds in duration, questions remain about how to efficiently handle visual and textual history when processing long sequences and how to minimize latency in such scenarios.
(3) Lack of Error Analysis: The paper lacks a detailed error analysis or discussion on failure cases, which would be useful to understand the model's limitations in edge cases.

---

> ### Author Rebuttal · Authors · 2025-07-31
>
> ### Rebuttal to Reviewer bqe1
>
>
> We sincerely thank the reviewer for the positive comments and constructive suggestions. In particular, we are delighted that you recognize our work as **“a novel task”**, that it **“advances multimodal conversational AI for real-time human-computer interactions”**, that it **“positions it as a promising candidate for future human-computer interactions”**, that the **“proposed dataset proves to be highly beneficial”**, and that our model **“generates high-quality multimodal listener responses.”** Your encouragement is greatly appreciated, and we will refine our manuscript carefully in accordance with your suggestions.
>
>
>
> >  ####  **Q1: Dependency on text modality.**
>
> A: Thank you for your comments.  We have conducted experiments  to evaluate whether relying on text would constrain the richness of non-verbal communication (see Table 2 of the main paper and table below).  That is, we build a baseline, namely **w.o Text** (i.e., Audio-visual LLM), which removes text modality from our method. We also provide another baseline, namely, LSTM-based method, employing widely-used network, i.e, a recurrent neural network to learn from only audio and visual modality.
>
>
>
>
> | Method                         | BERTScore F1 ↑ | Distinct-2 ↑ | LSE-D ↓ | UTMOSv2 ↑ | Non-verbal (FD) ↓ | Non-verbal (FVD) ↓ |
> |--------------------------------|----------------|--------------|---------|-----------|-------------------|--------------------|
> | LSTM                           | 0.716          | 0.000        | 9.72    | 1.21      | 6.51              | 320.92             |
> | w.o Text (Audio-visual LLM)    | 0.662          | 0.155        | 10.03   | 1.32      | 580.86            | 681.55             |
> | OmniResponse (Ours)            | **0.806**      | **0.882**    | **9.56**| **1.41**  | **15.46**         | **314.94**         |
>
>
> Our OmniResponse significantly outperforms the w.o Text baseline across all evaluated metrics, including non-verbal ones. These results demonstrate that introducing text as an intermediary modality significantly enhances the naturalness and realistic of non-verbal responses, as indicated by the FD and FVD scores. Similarly, our method outperforms the LSTM-based approach by a considerable margin.
>
> This superior performance is due to the combination of the intermediate text modality and our method's design:
>
> *  Text removes speaker-specific pitch and timbre, avoiding the negative effects by  noise or variability in the audio while  enabling the LLM to focus on intent understanding.
> *  Unlike raw speech, text does not affected by sampling rate or background audio noise, reducing the training difficulties for large-scale pre-training.
> * We introduce the Vision Projection module and Vision Decoder, and conduct Multimodal Context Modeling, allowing our model to learn from both visual modality itself and cross-modal correlations.
>
> We will add more discussions in the final version of our paper.
>
>
>
> >  ####  **Q2: How to efficiently handle visual and textual history when processing long sequences and how to minimize latency in such scenarios.**
>
>
> A: Thank you for your comments. Our method  handles long sequences and minimizes latency as follows:
> * We limit temporal inputs to a sliding window spanning the most recent $\Delta T$ = 2.2s (from time $\tau$ to $t-1$). All tokens preceding $\tau$ are distilled into a static textual summary, far cheaper than keeping them as full temporal tokens, shrinking the context length.
> * For the visual stream, only a short sequence of speaker‑listener face tokens is retained.
>
> * With these strategies, our system sustains 15.6 FPS (≈ 64 ms end-to-end delay for audio plus facial animation) on a single NVIDIA A100 80 GB GPU.
>
> To our knowledge, this is the first online multimodal conversational response generator. We outline additional efficiency improvements as future work, and the revised manuscript will detail the mechanisms that enable our pipeline to approach real-time operation.
>
> >  ####  **Q3: Discussion on Failure cases.**
>
> A: Thank you for your suggestion. Follow your suggestion, we conduct a detailed analysis of failure cases. First, we observe that when the speaker’s speech contains large background noise, our method would  wrongly generate  listener responses.  Second, our model rarely produces head rotations beyond approximately ±40°, and the resulting side-view frames can exhibit noticeable distortions. To quantify these effects, we extracted side-face and front-face frames from our generated videos and performed the following realism evaluation:
>
> | Category          | FVD ↓  |
> | ----------------- | ------ |
> | Side-face frames  | 478.31 |
> | Front-face frames | 312.69 |
> | All frames        | 314.94 |
>
> We will supplement these analyses and more failure case discussions in our revised version.
>
> >  ####  **Q4. Typo.**
>
> A: Thanks for your suggestion. We apologize for the typos and have corrected these typos.

---

> ### Author Response · Authors · 2025-08-07
>
> Dear Reviewer bqe1,
>
> Thank you for your careful and thoughtful review. We sincerely appreciate the time and effort you have devoted to reviewing our paper. Your insightful feedback has been invaluable in sharpening our arguments, broadening our experiments, and refining the clarity of our manuscript.
>
> We hope that our detailed explanations, additional results, failure cases provided, and expanded discussion have addressed your concerns fully. If you feel further experiments or information would be beneficial, we would be delighted to accommodate your suggestions.
>
> Should our revisions meet your expectations, we would be most grateful if you could reflect this in your final evaluation. In any case, we deeply appreciate the time and expertise you have dedicated to improving our work.
>
> Wishing you all the best and a pleasant remainder of your day.
>
> With sincere thanks,
>
> The Authors

---

> ### Comment · Area_Chair_PYCi · 2025-08-08
>
> Hi Reviewer bqe1,
>
> The author–reviewer discussion phase will close soon. If you have not yet participated, please take this opportunity to review the rebuttal, check how your comments have been addressed, and share any remaining concerns with the authors. Your engagement in these final days is key to ensuring a thorough review process.
>
> AC

---

### Comment · Area_Chair_PYCi · 2025-08-04
**Reminder: Review Rebuttal and Submit Final Justification**

Dear Reviewers,

As we approach the end of the author–reviewer discussion phase (**Aug 6, 11:59pm AoE**), I kindly remind you to read the author rebuttal carefully, especially any parts that address your specific comments. Please consider whether the response resolves your concerns, and if not, feel free to engage in further discussion with the authors while the window is still open.

Your timely participation is important to ensure a fair and constructive review process. If you feel your concerns have been sufficiently addressed, you may also submit your Final Justification and update your rating early. Thank you for your contributions.

Best,

ACs

---

### Note · Authors · 2025-08-14

We are deeply grateful to the ACs and reviewers for their generous time, thoughtful feedback, and constructive suggestions throughout the review process. Your feedback substantially improved clarity, evaluation rigor, and responsible-use framing.

1. We are encouraged by positive feedback on novelty, method design, dataset construction, and experimental results, such as:

   * **“novel task”** (bqe1, dTxZ, FU3z, c8bb) and “new task” (tzNS)
   * “**the first online model** to jointly process and generate synchronized streams of conversational human behavior” (dTxZ),  “*cleverly* circumvents the difficult problem” (c8bb),  "a *valuable baseline* for this new research direction" (tzNS).
   * “a **new, high-quality dataset**” (tzNS, dTxZ), “highly beneficial” (bqe1),  “*essential resource for the research community*” (FU3z).
   *  “*thorough evaluation*” (c8bb), “significantly outperforms baseline models” (dTxZ),  “generates high-quality multimodal listener responses” (bqe1).


2. We made every effort to address questions, concerns, and comments raised by all reviewers in the rebuttal and discussion. We briefly summarize below the status and outcomes of the rebuttal and discussion phases.

   * Reviewer **tzNS** recognized that our work *“provide a solid foundation for future studies"*, and stated that the reviewer had **raised the score, recommending acceptance**. Reviewers **c8bb, dTxZ, and FU3z** all had provided positive ratings before the author–reviewer discussion. They **maintained their positive ratings of 3/4 in all four evaluation criteria**, i.e., quality, clarity, significance, and originality during the discussion, where Reviewer c8bb explicitly indicated a decision to **raise the rating**.

   * Reviewer bqe1's main concerns are (i) reliance on text might constrain non-verbal richness, (ii) real-time applicability, and (iii) discussion of failure cases. For (i), Sec. 5.1 and Table 2 show that adding text modality enhances naturalness and realism rather than limiting expressiveness. For (ii), tzNS raised the same concern initially, but later stated it was addressed. For (iii), we conducted a detailed failure-case analysis. **We believe that we have addressed Reviewer bqe1's concern**.


We would like to once again express our sincere thanks to the AC for organizing the review process and to the reviewers for their constructive suggestions. We will incorporate these valuable suggestions into the final version to further improve our work.

---

### Decision · Program_Chairs · 2025-09-17

**Decision:**

Accept (poster)

**Comment:**

This submission initially received a mixed set of ratings. After the rebuttal discussion and subsequent discussions between reviewers and the area chair, several reviewers raised their scores in recognition of the authors' thorough and thoughtful responses. The authors addressed key concerns such as the reliance on the text modality, strategies for real-time processing, and failure case analysis, supported by additional experiments and clarifications. At this point, all reviewers are in favor of acceptance. After carefully reading the paper, the reviews, and the rebuttal, I agree with this decision. I recommend acceptance. I encourage the authors to further polish the camera-ready version by incorporating the reviewers' suggestions to improve clarity and completeness.